# Maximizing Lucerne (*Medicago sativa*) Pasture Intake of Dairy Cows: 1-the Effect of Pre-Grazing Pasture Height and Mixed Ration Level

**DOI:** 10.3390/ani10050860

**Published:** 2020-05-15

**Authors:** Kieran A. D. Ison, Marcelo A. Benvenutti, David G. Mayer, Simon Quigley, David G. Barber

**Affiliations:** 1School of Agriculture and Food Sciences, The University of Queensland, Gatton Campus, Lawes, QLD 4343, Australia; s.quigley@uq.edu.au; 2Queensland Department of Agriculture and Fisheries, Gatton Campus, Lawes, QLD 4343, Australia; marcelo.benvenutti@daf.qld.gov.au (M.A.B.); david.mayer@daf.qld.gov.au (D.G.M.); david.barber@daf.qld.gov.au (D.G.B.)

**Keywords:** lucerne, grazing management, grazing dynamics, *Medicago sativa*

## Abstract

**Simple Summary:**

Pre-grazing pasture height has significant impact on intake rate and intake in grazing dairy herds, and the ideal pre-grazing pasture height varies between pasture species. Defining the ideal pre-grazing pasture height of lucerne pasture has potential to significantly increase milk production of cows on sub-tropical partial mixed ration (PMR) dairies in Australia. Pasture intake of dairy cows was highest when the pre-grazing pasture height of lucerne was 39 cm and a proportion of the pasture remained un-grazed, irrespective of the amount of mixed ration offered.

**Abstract:**

The effect of lucerne (*Medicago sativa* L.) pre-grazing pasture height on pasture intake and milk production was investigated in a sub-tropical partial mixed ration (PMR) dairy system in south-east Queensland, Australia. The experiment involved a 26-day adaptation period followed by an eight-day measurement period during April and May 2018. Twenty-four multiparous Holstein-Friesian dairy cows were offered a mixed ration at either 7 (low) or 14 (high) kg dry matter (DM)/cow/day and allocated pastures at pre-grazing heights ranging from 23 to 39 cm. The targeted pasture intake was 14 and 7 kg DM/cow/day for cows offered the low and high mixed ration allowances respectively, with a total intake target of 21 kg DM/cow/day. Pasture structure did not limit pasture intake as the all groups left at least 12% of the allocated area ungrazed, and therefore could selectively graze pasture. There was no significant difference in intake between mixed ration levels, however intake had a positive linear relationship with pre-grazing pasture height. For every one cm increase in pasture height, intake increased by 0.3 kg DM/cow/day. Using a grazing strategy that ensures the some pasture remains ungrazed and the pre-grazing height of lucerne is approximately 39 cm above ground level will maximise pasture intake in sub-tropical PMR dairy systems.

## 1. Introduction

Pastures are a highly important source of forage on many commercial dairy farms throughout the sub-tropical regions of Australia [1,2]. Recently, climatic conditions in these regions of Australia have become less predictable and consequently pasture productivity has declined and become less consistent [1]. Consequently, farmers have adapted new strategies to utilise conserved forages to maintain production when pasture productivity is limited. These systems incorporate both high quality, low cost pastures and a mixed ration usually containing starch and protein based forages combined with concentrates and minerals, referred to as partial mixed rations (PMR) [2]. These systems are relatively new within the sub-tropical regions of Australia and the interactions between pasture and mixed ration within the rumen, and the consequent effects on intake and production are not well understood when tropical forages are fed [2,3]. 

Increasing intake within a pasture based or total mixed ration system can be achieved by altering the quantity or quality of either the pasture or mixed ration [4,5]; however, there is limited knowledge on the effect in PMR systems. It is well understood that pasture utilisation and intake can be manipulated by pasture allocation and pre-grazing pasture height [6]. Therefore, manipulating either one of these parameters through grazing management strategies is likely to have significant effects on intake and productivity.

Lucerne (*Medicago sativa*) pasture has a moderate reliance within the sub-tropical region of Australia (extending from north-east New South Wales to far-north Queensland). With better understanding of management practices to optimize production, lucerne pasture may become more prevalent due to its drought resistance, tolerance to grazing and high quality [7]. Within intensive livestock systems, defoliation intensity (area grazed and bite depth) and pasture intake is driven by both pasture structure and allocation. Recently, Ison et al. [8] found that high total intakes could be achieved by sub-tropical PMR dairy herds when offered large allocations of lucerne pasture. This study indicated that high pasture intake and milk yield were only achieved when approximately 5% of pasture within an allocation remained un-grazed. Diet quality was also maximised when some pasture remained un-grazed as cows selectively removed the higher quality top leafy stratum of the pasture sward [8,9]. These results led to the development of a new grazing management strategy, termed proportion of uncontaminated, un-grazed pasture (PUP). The PUP grazing strategy aims to offer allocations of pasture to the lactating herd to ensure that a small proportion of pasture remains un-grazed, including around the faecal patches and thus, cows are only removing the top leafy stratum. This results in maximum pasture intake and diet quality with potential to increase production and margin over feed cost as it can minimize the use of expensive supplements or mixed ration ingredients. The residual pasture is then either utilised by non-milking cows or removed by mechanical methods [8,10]. This previous study identified how pasture allocation and therefore the PUP strategy drives lucerne pasture intake in a sub-tropical PMR dairy system, however it did not identify the optimum pre-grazing pasture height that maximizes intake. The current study looked at the effect of pre-grazing pasture height on pasture intake under the PUP grazing strategy.

The pre-grazing pasture height as a driver of pasture intake has been well studied in some pasture species [6,11] but not for lucerne within sub-tropical PMR dairy systems. Typically, as plant height increases the structure of the sward changes due to the variation in the vertical distribution of the leaf and stem [12]. This variation dictates the bite mass as the cows selectively remove the top leafy stratum, and as a consequence, results in changes in intake [12,13]. Previous studies [6,11] found a quadratic relationship between pre-grazing pasture height and pasture intake rate. Bite mass and intake rate initially increased with pre-grazing pasture height to a maximum level and then decreased due to the reduction in bite mass. Therefore, using pre-grazing pasture height to define an ideal stage for grazing to maximize pasture intake would have a significant impact on intake and the cost of production on commercial dairy farms.

The objective of this study was to identify the optimum pre-grazing pasture height of lucerne pasture to maximise intake when pasture allocation is high and not limiting intake [14] in a sub-tropical PMR dairy system. The hypotheses tested were: (1) pasture intake would respond in a quadratic fashion with increasing pre-grazing pasture height; (2) mixed ration allocation would not affect total intake irrespective of pre-grazing pasture height and; (3) milk production would increase as total intake increased due to a higher total metabolisable energy intake. 

## 2. Materials and Methods

The study took place at the Gatton Research Dairy, south-east Queensland, Australia (−27.552, 152.333), and consisted of a 26-day adaptation period (17 April–11 May 2018) followed by an eight-day measurement period (12–19 May 2018). The study was conducted in agreement with the guidelines of the Australian Code of Practice for the Care and Use of Animals for Scientific Purposes (National Health and Medical Research Council 2013) and was approved by the Department of Agriculture and Fisheries Animal Ethics Committee (reference number SA 2018/02/632). 

### 2.1. Experimental Design

Twenty-four multiparous Holstein-Friesian lactating dairy cows (*Bos taurus*) were randomly allocated to six mixed ration-grazing treatment combinations with two grazing replicates per treatment (*n* =2 cows/replicate). The cows were randomly allocated to treatment groups based on a principal components summarization of days in milk (166 ± 50), milk yield (27.9 ± 3.6 L/cow/day), milk fat concentration (3.68 ± 0.30%), milk protein concentration (3.10 ± 0.18%), somatic cell count (80,000 ± 35,000 cells/mL), liveweight (610 ± 49 kg), body condition score (4.7 ± 0.3; 1–8 scale) and parity (2.5 ± 0.9). The experimental treatments consisted of a factorial combination of two mixed ration intakes [7 (low) and 14 (high) kg dry matter (DM)/cow/day], and three targeted levels of pre-grazing height (short: 25 cm, medium: 35 cm and tall: 50 cm) of a pure lucerne pasture. The target pasture intakes were 14 and 7 kg DM/cow/day for low and high mixed ration treatments respectively, representing 33% and 67% of the targeted total diet intake of 21 kg DM/cow/day. Cows were offered the mixed ration in individual groups on a typical feedlot feeding structure (feed pad) following the AM milking at 0900 h until the afternoon milking at 1500 h. After PM milking (1700 h), cows were moved to their allocated lucerne pasture strip to graze overnight until the following AM milking at 0630 h. The two mixed ration diets were formulated using the Nittany Cow ration formulation program (NittanyCow, ‘Dairy Ration Evaluator’ Software, Mifflinburg PA, USA). Diets were balanced to meet the metabolic requirements of cows producing an average of 25 L/cow/day when combined with the expected nutrient intake of either 7 or 14 kg of DM of the lucerne pasture for high and low mixed ration groups respectively (Table 1). Note that actual pasture intake could not have been predicted before the experiment as it was not known; therefore formulating a diet for each pasture height treatment was not possible. Daily mixed ration refusals were removed and weighed to calculate the average daily mixed ration intake for each group of two cows. Pasture intake was calculated as an average for each group of two cows, detailed in Section 2.2.

### 2.2. Pasture Allocation and Nutritive Value

Pasture allocation, intake and quality were determined by calculating the top-down vertical distribution of DM and chemical composition of the lucerne pasture. This was determined using a random, stratified double-sampling method adapted from Benvenutti et al. [15] and used by Ison et al. [8] to estimate and explain nutrient intake from lucerne pastures. Twenty-two pasture samples (ranging in height from 20 to 55 cm) were cut 5 cm above ground level, and then further cut into four equal vertical strata. Pasture samples were taken at the beginning, middle and end of each experiment period. Within each stratum DM was determined by drying samples to 60 °C. Sub-samples from within each stratum were analysed at Dairy One Forage Lab (Ithaca, NY, USA) to estimate crude protein (CP), % DM [16]; neutral detergent fibre (NDF, A-amylase was not used to determine NDF) and acid detergent fibre (ADF), % DM [17]; lignin, starch and sugar % DM [18]. All values include ash. Concentrations of metabolisable energy (ME) were calculated using the following formula [19]:ME (MJ/kg DM) = (((1.01 × (0.04409 × TDN)) − 0.45) × 4.184(1)
where TDN is total digestible nutrient (%). TDN was estimated using the following formula [19]:TDN = 5.31 + 0.412 CP % + 1.444 Ether Extract % + 0.937 Nitrogen Free Extract %(2)

These data were then used to develop calibration equations of the top-down vertical distribution of DM and chemical composition of the pasture as described by Ison et al. [8]. Pasture was allocated to each treatment group using the estimated intake from grazing the top leafy stratum using the calibration equations. The area required to reach the target pasture intake was doubled to ensure pasture allocation did not limit pasture intake. 

### 2.3. Defoliation Intensity and Pasture Intake

Defoliation intensity was defined by calculating the vertical and horizontal utilisation of the pasture, using height measurements taken at fixed assessment points along transects within each grazing strip. Transects where taken across the grazing strip at 2 m intervals, and measurements of pasture height were recorded every 1 m along each transect. On average, two transects were taken for each paddock and were 50 m in length, with a total of 100 measurement points taken in each individual strip per day. Using the vertical distribution of DM and chemical composition of the pasture outlined in Section 2.2, pasture intake and quality were calculated as described by Ison et al. [8] with an estimated error of ±18%. Pasture consumed at each fixed assessment point was calculated from the measured grazing depth and the top-down vertical distribution of DM from Section 2.2. Pasture intake was calculated as the average pasture consumed from the assessment points excluding the trampled and contaminated points. Trampled and contaminated points were visually assessed at each measurement point within each grazing strip and recorded as ‘ungrazable’ areas as cows cannot graze trampled pasture and avoid grazing areas contaminated with feces [15]. These values were subtracted from the total area allocated to determine the maximum grazable area within each strip. The combined horizontal and vertical utilisation of the grazed pasture was used to calculate the average intake for each treatment group every day. 

### 2.4. Milk Production

Milk yield was measured at each milking time (0700 and 1500) for individual cows by using automatic flow meters (Westfalia, ‘Dairy Plan’ Software, Düsseldorf, Germany), with milk samples taken for individual cows at both milking times throughout the measurement period, analysed for fat and protein (Siliker Australia, Eagle Farm, Brisbane, QD, Australia). Energy-corrected milk standardized to 4.0% fat and 3.3% protein, was calculated using the following formula [20]:Energy corrected milk (kg/cow/day) = milk yield (kg) × (376 × fat% + 209 × protein% + 948)/3138(3)

### 2.5. Statistical Analysis

All analyses were conducted using the GenStat^®^ (18th Edition, VSN International Ltd., Hemel Hempstead, Hertfordshire, England) software package. Multiple linear regressions were used to develop the calibration equations for pasture mass and forage quality parameters. The backward (step-down) selection method was used, and explanatory variables were removed if *p* > 0.05. For the response (Y) variables, the independent experimental units were the replicates (each with two cows), with the replicates being fitted as random effects in the analyses. General linear models were used to assess the effects of the mixed ration (as a factor of two levels) and pre-grazing height (as a linear contrast; curvature of these responses was tested but none were significant), including their interaction, on the response variables. 

## 3. Results

### 3.1. Defoliation Intensity and Pasture Structure

The lucerne pasture had a typical structure with a top leafy stratum and a bottom-stemmy stratum [10]. The average pre-grazing pasture height for each treatment group ranged from 24 to 40 cm (Table 2). The variability in pre-grazing pasture height between pasture height categories was likely caused by variability in soil moisture and nutrient availability on the area assigned to each treatment group. This combined with a sudden change in temperature caused the pasture growth to decline and the target pre-grazing heights described in Section 2.1 were not achieved for the tall treatments. Therefore, for all analyses, pre-grazing pasture height (cm) was taken as the explanatory variable.

Pasture offered (kg DM/cow/day) had a significant positive linear relationship with pre-grazing pasture height (*p* < 0.001) and averaged 45.4 and 22.0 kg DM/cow/day for low and high mixed ration levels respectively (Table 3). The average grazing depth did not have a significant relationship with pre-grazing pasture height, and averaged 11.7 and 13.5 cm for low and high mixed ration levels respectively (Table 3). The area grazed had a significant positive linear relationship with pre-grazing pasture height, and was not different between mixed ration levels (Table 2). 

### 3.2. Dry Matter Intake and Diet Quality

Mixed ration intake did not have a significant relationship with pre-grazing pasture height (*p* > 0.05), but tended to be higher (*p* = 0.099) for cows with a high mixed ration intake (11.5 kg DM/cow/day) than the cows with a low mixed ration intake (7.3 kg DM/cow/day) (data not presented). Cows in high and low mixed ration levels were allocated at least 15 and 34 kg DM pasture/cow/day respectively (Table 2). Pasture intake had a significant positive linear relationship with pre-grazing pasture height (*p* < 0.05) and it was significantly different between mixed ration levels (*p* < 0.05) (Figure 1). Total intake had a significant positive linear relationship with pre-grazing pasture height but was not significantly different between high and low mixed ration levels (Figure 2).

The ME intake from the lucerne pasture, and the NDF and ADF content of the lucerne had a significant positive relationship with pre-grazing pasture heights (*p* < 0.05) (Table 4). The CP, lignin, starch and sugar content of the consumed lucerne pasture had a significant negative relationship with pre-grazing pasture heights (Table 3). Pasture ME intake, and CP, starch and sugar content was high for cows with a low mixed ration intake (Table 3). Pasture NDF, ADF and lignin content was lower for cows with a low mixed ration intake (Table 3). 

The proportion of mixed ration in the entire diet had a significant negative relationship with pre-grazing pasture height and was lower for cows with a low mixed ration intake. Therefore, the proportion of pasture in the entire diet had a significant positive relationship with pre-grazing pasture height and was higher for the low mixed ration level (*p* < 0.05) (Table 4). The nutrient content (ME, CP, NDF, ADF, lignin, starch and sugars) of the total diet is shown in Table 5. Mixed ration intake is not shown as there were no significant differences in intake within mixed ration levels, and therefore any relationships between total diet quality and pre-grazing pasture height are due to differences in pasture intake. Total diet ME intake (MJ/cow/day) had a significant positive relationship with pre-grazing pasture height (*p* < 0.01) and was not different between mixed ration levels (*p* > 0.05) (Table 4). CP and sugars of the total diet had significant positive relationships with pre-grazing pasture height (*p* < 0.01) and were significantly higher for the low mixed ration level (*p* < 0.001) (Table 4). NDF and ADF had significant relationships with pre-grazing pasture height (*p* < 0.05) and were higher for the high mixed ration level (*p* < 0.05 (Table 4). Lignin content of the total diet had a significant positive relationship with pre-grazing pasture height (*p* < 0.01) and was significantly lower for the high mixed ration level (*p* < 0.05) (Table 4). Starch content of the total diet had a significant negative relationship with pre-grazing pasture height (*p* < 0.01) and was significantly higher for the high mixed ration level (*p* < 0.001) (Table 4).

### 3.3. Milk Production

Milk yield (L/cow/day) (Figure 3), milk protein (%) (Figure 5) and energy-corrected milk yield (kg/cow/day) (Figure 6) did not have a significant relationship with pre-grazing pasture height and were not significantly different between mixed ration levels. Milk yield (L/cow/day), milk protein (%) and energy-corrected milk yield (kg/cow/day) averaged 27.3 ± 0.53, 3.2 ± 0.15 and 25.0 ± 0.59 respectively. Milk fat (%) (Figure 4) did not have a significant relationship with pre-grazing pasture height, however was significantly higher for the low mixed ration level (*p* < 0.001). Milk fat (%) averaged 3.78 ± 0.25 and 2.92 ± 0.23 for low and high mixed ration levels, respectively.

## 4. Discussion

This study examined the effects of increasing pre-grazing pasture heights under the PUP grazing strategy and two mixed ration levels on defoliation intensity, intake, and milk production and composition of lactating dairy cows. The amount of mixed ration offered and targeted total intakes within this study are consistent with those of commercial sub-tropical PMR dairy systems [21].

The insignificant difference of grazing depth between treatments indicates that the defoliation dynamics were similar for all groups. The average grazing depths of 13 cm for both mixed ration levels are similar to results from Ison et al. [8], which showed that cows preferred to graze to a consistent depth in lucerne pastures, comprised mainly of leafy material, removing approximately 950 kg DM/ha at an average depth of 15 cm. These data suggest that cows preferentially graze the top leafy stratum of lucerne pasture irrespective of pre-grazing pasture height. For each of these studies, grazing depth ranged from ~28 to 57% and ~24 to 28% of the sward height respectively. Previous studies on various vegetative pastures have found that bite depth was ~50% of sward height [13,22,23]. However, Benvenutti et al. [15,24] measured grazing depths of up to 65% to 86% of the pasture height for *Cynodon* spp. or *Axonopus catharinensis*. The large variation in grazing depth is likely due to the large variability of the depth of the top leafy stratum, which has significant effects on grazing depth [8,15,24]. The variability of the depth of the top leafy stratum of pastures between species, suggests that large pasture allocations are used and cows can selectively graze the top leafy stratum, grazing depth may be better estimated from the depth of the top leafy stratum rather than a proportion of the pasture height.

The higher intake of taller pastures is consistent with Laca et al. [23] who showed that intake was higher in cows grazing taller lucerne pasture due to a higher bulk density and consequently larger bite mass. Furthermore, the area grazed increased as pre-grazing pasture height increased which further explains the higher pasture intakes achieved with taller pastures. The reason for the larger grazing area is unclear. It is possible that cows preferentially grazed the taller pastures, however this relationship is complex and not well understood [25,26]. Higher pasture intake explains the increase in total intake as there was no difference in mixed ration intake between the two mixed ration levels. However, mixed ration intake in the high mixed ration groups was on average 2.5 kg less than the targeted 14 kg DM. This may have been limited caused by insufficient time for the cows to consume the mixed ration, and consequently why targeted total intakes were not achieved. Conversely, groups on low mixed ration allocations consumed all offered mixed ration, and only reached a maximum pasture intake of 12 kg DM, 2 kg less than the targeted pasture intake. It is possible that cows had insufficient time to consume either the mixed ration or pasture for high and low mixed ration levels respectively, limiting total intake.

We hypothesized that pasture intake would have a quadratic relationship with pre-grazing pasture height, increasing as height increased and declining at extreme pasture heights as found in previous studies using grass species [6,11]. These previous studies found that bite mass and intake rate initially increased with pasture height to a maximum level and then decreased due to the reduction in bite mass. Bite mass was lower in tall pastures due to their lower bulk density. However we did not observe this quadratic relationship as intake increased linearly with increasing pasture height. Two reasons could explain the different responses. The first reason is that our tall lucerne pastures were not mature enough for the animals to become selective and reduce bite mass while grazing the leaves and avoiding the stems. Therefore, further studies should be conducted to look at the effect of the interaction between pasture height and stage of maturity on pasture intake and milk production. The second reason relates to the differences in plant structure between grasses and legumes. The bulk density of the top leafy stratum decreases with pasture height for grasses but increases for lucerne [22]. This may explain why pasture intake did not decrease in our tall lucerne pastures. The results in the current study are similar to [27], where beef cattle had highest intakes on the tallest and largest allocations of lucerne pastures. In this current study, the highest intake irrespective of mixed ration level was achieved when the pre-grazing pasture height was tallest, 39 cm above ground level. Therefore, within sub-tropical PMR dairy systems, lucerne pasture should be allocated to ensure PUP does not reach 0% [8] (also see the companion paper in this special issue [28]), at a pre-grazing height not less than 39 cm to maximise intake when the mixed ration allocation ranges from 33% to 67% of the total diet. 

The results support our second hypothesis that the mixed ration allocation would not affect total intake irrespective of pasture height. Low mixed ration intake resulted in higher pasture intake, and vice versa, resulting in similar total intakes. Previous studies also found that pasture intake increased with decreasing mixed ration allocations [14].

The quality of the consumed pasture varied as pasture height increased and varied between mixed ration levels. Increasing maturity of the plant as pasture height increases causing changes in nutrient composition of both the leaf and stem tissues within legume pastures [12]. This explains the variability of pasture quality as PGPH increased. Despite the relationship between PGPH and nutrient content being statistically significant, the slopes of the relationships were rather low and probably biologically insignificant. Therefore, the observed increase in energy and nutrient intake with increasing PGPH was mainly driven by increasing pasture intake rather than the changes in nutrient content of the consumed pasture. The variability of consumed pasture quality between mixed ration levels may be explained by the small difference in grazing depth between mixed ration levels. However, this relationship was not significant and further investigation is required. The variability in total diet quality was largely driven by changes in pasture intake and the consequent proportions of mixed ration within the whole diet. The increasing intake and energy content of the pasture as pasture height increased explains the increase in total DM and energy intake. The variability of CP, NDF, ADF, lignin, starch and sugar (%) of the total diet was likely driven by the variability of the pasture combined with the variable proportion of mixed ration in the whole diet. The CP of the total diet increased significantly as the proportion of pasture in the total diet increased. Conversely, starch declined significantly. These trends are expected as the mixed ration was formulated to be high in energy and starch to provide a total diet energy close to requirements when balanced against targeted pasture intakes. 

We hypothesized that milk production would increase as intake increased, however milk yield (L/cow/day), milk protein (%) and energy-corrected milk yield (kg/cow/day) did not have a significant relationship with pre-grazing pasture height and were not different between mixed ration levels (Figure 3, Figure 5 and Figure 6). These outcomes are in contrast to previous PMR studies where milk yield and milk components typically increase as intake and diet quality increases [29,30]. Cows with lower total intake possibly mobilized body reserves to ameliorate the negative impacts of lower energy intake on milk yield. Body condition score, liveweight and blood plasma metabolites were recorded in this study to provide an indication of body reserve utilisation, however differences between treatments were not detected in this study (data not shown). Despite the observed significant effect of pasture height on pasture intake and total diet intake, we did not detect the effect of pasture height on milk production. This is possibly explained by the short period of the study. A long-term study is necessary to confirm these results. The difference in milk fat (%) between mixed ration levels is likely caused by the higher proportion of starch with the high mixed ration intake. The average proportion of starch in the high mixed ration diets was 28.9% in contrast to the 20.2% maximum in the low mixed ration groups. National Research Council [19] recommends starch comprises 22% to 25% of the diet to avoid milk fat suppression in lactating dairy cows. Various studies have shown that diet is a major factor in milk fat suppression [30,31], specifically diets high in starch and lipids [32]. Typically in these diets, rumen pH declines, altering the microbial population within the rumen where biohydrogenation increases and consequently milk fat declines [33]. 

## 5. Conclusions

Pasture and total intake increased as pasture height increased and total intake was not different between mixed ration levels. The pre-grazing pasture height of lucerne within this study did not reach extreme levels, and a typical quadratic response of intake to pasture height was not observed. Milk production did not decline as total intake declined, possibly due to the short duration of the study. This study showed that lucerne pasture should be grazed when pre-grazing pasture height is at least 39 cm above ground level, and allocated to ensure an area remains un-grazed (PUP) to maximise intake in PMR dairy systems. Targeted total intakes were not achieved, likely due to insufficient time to consume either the mixed ration or pasture. Further investigation is required to define the minimum time required to achieve targeted pasture and mixed intakes within PMR dairy systems. This study also highlights that understanding the actual contribution of pasture to the diet has significant effects on production. Although there were no significant effects on intake and milk yield, the current study indicated that the high starch intakes achieved when pasture intake was lower than expected had detrimental effects on milk fat composition. This is likely to exacerbate over long periods, with potentially significant herd health and economic impacts. Further investigation is required to define an ideal post-grazing target for lucerne pastures at varying mixed ration allocations to maximise intake. This was investigated in the second experiment presented in the companion paper published in this special issue. 

## Figures and Tables

**Figure 1 animals-10-00860-f001:**
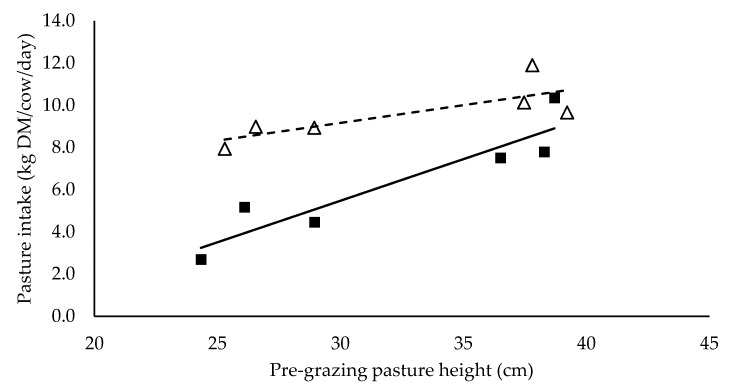
The relationship between pasture intake and pre-grazing pasture height (cm) for high (■) and low (Δ) mixed ration levels. Lines were fitted for high (solid) and low (dashed) mixed ration levels.

**Figure 2 animals-10-00860-f002:**
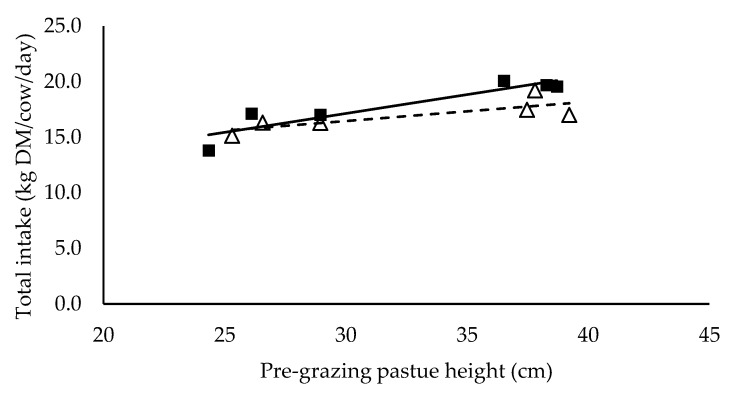
The relationship between total intake and pre-grazing pasture height (cm) for high (■) and low (Δ) mixed ration levels. Lines were fitted for high (solid) and low (dashed) mixed ration levels.

**Figure 3 animals-10-00860-f003:**
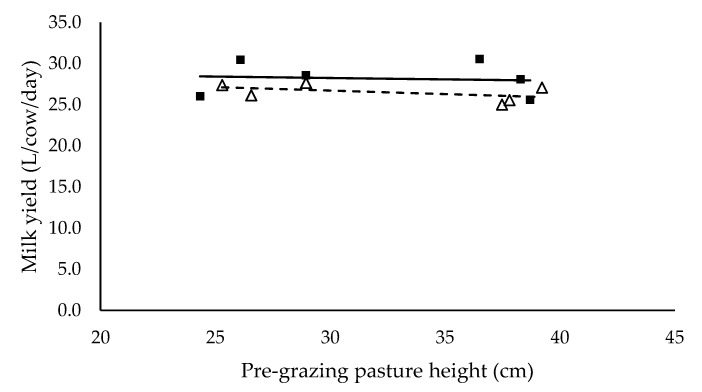
The relationship between milk yield and pre-grazing pasture height (cm) for high (■) and low (Δ) mixed ration levels. Lines were fitted for high (solid) and low (dashed) mixed ration levels.

**Figure 4 animals-10-00860-f004:**
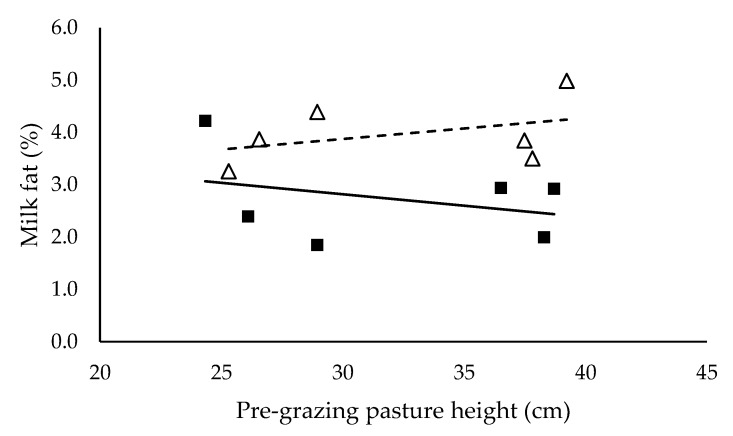
The relationship between milk fat and pre-grazing pasture height (cm) for high (■) and low (Δ) mixed ration levels. Lines were fitted for high (solid) and low (dashed) mixed ration levels.

**Figure 5 animals-10-00860-f005:**
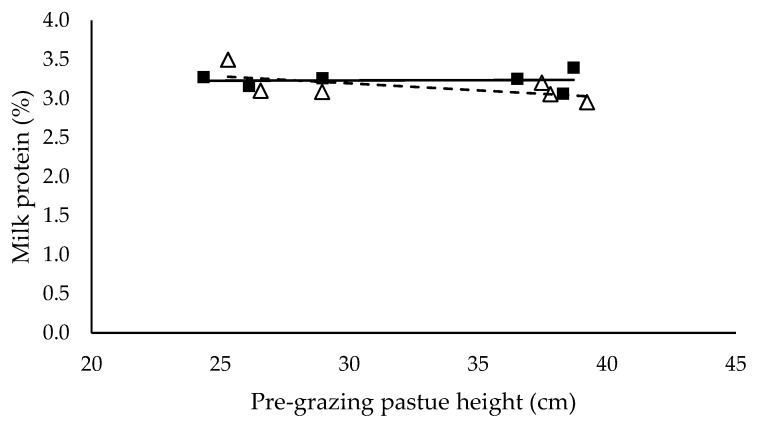
The relationship between milk protein and pre-grazing pasture height (cm) for high (■) and low (Δ) mixed ration levels. Lines were fitted for high (solid) and low (dashed) mixed ration levels.

**Figure 6 animals-10-00860-f006:**
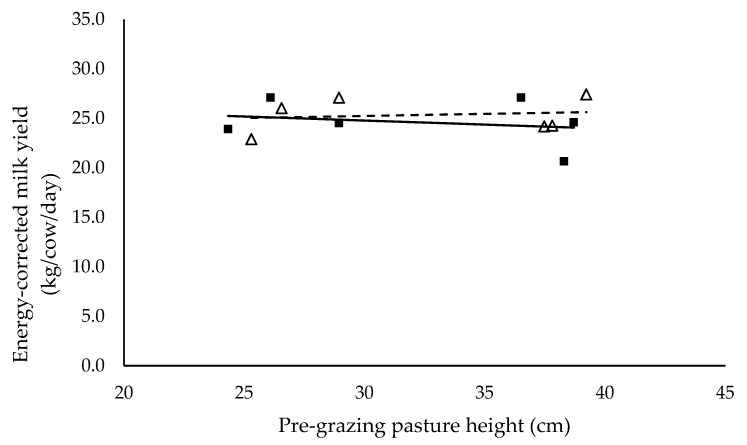
The relationship between energy-corrected milk yield and pre-grazing pasture height (cm) for high (■) and low (Δ) mixed ration levels. Lines were fitted for high (solid) and low (dashed) mixed ration levels.

**Table 1 animals-10-00860-t001:** Diet ingredient composition for the high and low mixed rations treatments with the targeted pasture intakes.

Ingredient (kg DM/cow/day)	High	Low
Corn silage	6.71	4.15
Barley grain	3.15	1.22
Wheat grain	1.89	0.75
Canola meal	0.9	0
Lucerne hay	0.9	0.45
Mineral mix	0.44	0.44
Total mixed ration	13.99	7.01
Target pasture intake	7.0	14.0
Target total intake	21.0	21.0

**Table 2 animals-10-00860-t002:** Relationship between pre-grazing pasture height and grazing depth and average area grazed by cows with high and low mixed ration level.

Mixed Ration Level	Target Pre-Grazing Pasture Height (cm)	Actual Pre-Grazing Pasture Height (cm)	Pasture Offered (kg DM/ha)	Pasture Offered (kg DM/cow/day)	Grazing Depth (cm)	Average Area Grazed (%)
High	25	24.34	956.21	15.68	14.02	37.41
High	25	26.10	1047.34	17.07	11.71	48.27
High	35	28.95	1034.89	17.53	12.27	51.29
High	50	36.51	1492.56	26.56	16.57	49.67
High	50	38.29	1579.60	28.61	12.44	77.35
High	35	38.70	1488.57	26.64	14.13	87.47
Low	25	25.30	1007.36	34.60	10.15	48.85
Low	25	26.56	1071.19	38.29	12.73	38.10
Low	35	28.94	1034.58	36.11	11.33	54.50
Low	50	37.47	1539.45	55.51	10.51	53.00
Low	50	37.80	1555.71	54.54	11.68	60.05
Low	35	39.22	1512.44	53.23	13.75	51.49
			R^2^	0.98	0.38	0.67
			PMR effect ^+^	***		
			Slope (low)	1.54 ***	0.071	0.71
			Slope (high)	0.89 ***	0.105	2.49 **

Data presented is the measured data for each replicate. ^+^ From the analyses, highest significance level of either the mixed ration main effect or its interaction with pre-grazing pasture height. Significance indicated by ** *p* < 0.01 and *** *p* < 0.001.

**Table 3 animals-10-00860-t003:** The relationship between pre-grazing pasture height (cm) and metabolisable energy (MJ/cow/day) intake and nutrient content (% DM) of the consumed lucerne pasture for each treatment group.

Mixed Ration Level	Pre-Grazing Pasture Height	Metabolisable Energy	Crude Protein	Neutral Detergent Fibre	Acid Detergent Fibre	Lignin	Starch	Sugars
High	24.34	33.0	34.83	16.74	16.20	5.66	5.86	8.78
High	26.10	62.9	34.08	17.77	17.05	5.73	5.40	8.65
High	28.95	54.6	33.95	17.66	16.78	5.58	5.49	8.66
High	36.51	92.7	32.85	18.62	17.20	5.36	5.20	8.54
High	38.29	96.6	32.72	18.64	17.11	5.28	5.25	8.53
High	38.70	127.0	31.97	19.81	18.16	5.43	4.82	8.38
Low	25.30	97.2	34.46	17.24	16.60	5.69	5.63	8.72
Low	26.56	110.2	34.32	17.34	16.62	5.65	5.60	8.70
Low	28.94	109.7	33.92	17.71	16.82	5.59	5.47	8.66
Low	37.47	126.7	33.36	17.69	16.29	5.18	5.60	8.66
Low	37.80	148.6	33.16	17.99	16.54	5.21	5.49	8.62
Low	39.22	121.3	33.25	17.68	16.17	5.10	5.61	8.66
	R^2^	0.88	0.94	0.84	0.71	0.95	0.76	0.84
	Mixed ration effect ^+^	*	*	*	*	*	*	*
	Slope (low)	0.0225 ***	−0.088 ***	0.035	−0.0283	−0.0420 ***	−0.0013	−0.0046
	Slope (high)	0.0080 *	−0.159 ***	0.149 ***	0.0764 *	−0.0256 ***	−0.0457 **	−0.0196 ***

Data presented is the measured data for each replicate. ^+^ From the analyses, highest significance level of either the mixed ration main effect or its interaction with pre-grazing pasture height. Significance indicated by * *p* < 0.05, ** *p* < 0.01 and *** *p* < 0.001.

**Table 4 animals-10-00860-t004:** The relationship between pre-grazing pasture height (cm) and metabolisable energy (MJ/cow/day) intake and nutrient content (% DM) of the total diet for each treatment group.

Mixed Ration Level	Pre-grazing Pasture Height	Mixed Ration ^a^	Pasture ^a^	Metabolisable Energy	Crude Protein	Neutral Detergent Fibre	Acid Detergent Fibre	Lignin	Starch	Sugars
High	24.34	80.6	19.4	139.90	19.87	21.75	15.53	3.49	28.88	3.58
High	26.10	70.2	29.8	182.23	21.67	21.28	15.86	3.80	25.93	4.30
High	28.95	74.8	25.2	170.94	20.78	21.52	15.76	3.61	27.26	4.00
High	36.51	63.1	36.9	211.47	22.47	21.26	16.09	3.86	23.78	4.69
High	38.29	60.4	39.6	212.99	22.84	21.12	16.02	3.89	23.03	4.84
High	38.70	47.5	52.5	217.77	24.56	21.25	16.83	4.28	18.97	5.55
Low	25.30	49.3	50.7	151.98	24.38	19.93	16.27	4.31	20.23	5.60
Low	26.56	45.9	54.1	178.49	24.99	19.85	16.24	4.37	19.18	5.79
Low	28.94	47.3	52.7	152.13	24.45	20.13	16.53	4.32	19.46	5.66
Low	37.47	42.8	57.2	191.22	25.09	19.86	16.13	4.21	18.23	5.95
Low	37.80	38.7	61.3	209.86	25.79	19.67	16.20	4.31	17.07	6.18
Low	39.22	44.1	55.9	184.63	24.77	19.95	16.03	4.13	18.61	5.87
	R^2^	0.92	0.92	0.78	0.90	0.97	0.63	0.85	0.91	0.93
	Mixed ration effect ^+^	***	***		***	***	*	***	***	***
	Slope (low)	−0.0048	0.0048	2.81 *	0.00048	−0.00009	−0.00017	−0.00011	−0.0014	0.00026
	Slope (high)	−0.0161 **	0.0161 **	4.41 **	0.00224 **	−0.00027 *	0.00054 *	0.00034 **	−0.0048 **	0.00096 ***

Data presented is the measured data for each replicate. **^a^** Indicates percentage (%) of total diet. ^+^ From the analyses, highest significance level of either the mixed ration main effect or its interaction with pre-grazing pasture height. Significance indicated by * *p* < 0.05, ** *p* < 0.01 and *** *p* < 0.001.

**Table 5 animals-10-00860-t005:** The summary statistics of the linear relationships between all response variables in Figure 1, Figure 2, Figure 3, Figure 4, Figure 5 and Figure 6 and the explanatory variable, pre-grazing pasture height (cm).

Y Variate	R^2^	High Intercept	High Slope	Low Intercept	Low Slope
Pasture intake (kg DM/cow/day)	0.89	−6.35	0.394 ***	4.12	0.168 *
Total intake (kg DM/cow/day)	0.80	6.93	0.340 **	11.28	0.173 *
Milk yield (L/cow/day)	0.69	29.5	0.0287	30.4	−0.0296
Energy-corrected milk yield (kg/cow/day)	0.53	1.74	0.0131	3.59	−0.0005
Milk fat (%)	0.25	2.92	0.0042	2.88	0.0047 *
Milk protein (%)	0.46	20.9	0.718	28.4	−0.0024

Significance of slopes indicated by * *p* < 0.05, ** *p* < 0.01 and *** *p* < 0.001.

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
