# Peer review of "Maximizing Lucerne (Medicago sativa) Pasture Intake of Dairy Cows: 1-the Effect of Pre-Grazing Pasture Height and Mixed Ration Level"

_animals, 2020, doi:10.3390/ani10050860_

Round 1
Reviewer 1 Report
The revised version is prefer than the previous one.
The authors appropriately responded to my concerns; however, I suggest several minor concerns in your revision.
Line 58: Lucerne (Medicago sativa) pasture instead of "Lucerne pasture".
Lines 60–61: What is the subject of this sentence? "Lucerne pasture may become..."?
Line 64: Remove "(Medicago sativa)".
Lines 138–140: Please add brief description of Bevenutti's and Ison's methods to understand how to determine pasture intake.
Line 144: What is "mincing energy"?
Line 191–193: I believe that the replicate should be included in the GLM as a random factor.
Lines 235–259: Please double-check the position of Tables in the text. For example, the relationship between pre-grazing height and pasture content was shown in Table 3, but you referred Table 4 in line 236.
Although, the most of the relationship between pre-grazing height and nutrient content were statistically significant, the slopes were rather low. For example, if the pasture height increase 10 cm, the CP content in the pasture decreases only 0.88%. Is there substantial meaning in this relationship? Even if the P-values show significance, you should consider its biological meaning.
Table 3 and 4: "ME intake" should be included in the titles, since all other items such as CP and NDF mean "content" in lucerne and total diet.
Author Response
Line 58: Lucerne (Medicago sativa) pasture instead of "Lucerne pasture".
- ‘Lucerne (Medicago sativa) pasture’…. corrected in text (line 1 and 58).
Lines 60–61: What is the subject of this sentence? "Lucerne pasture may become..."?
- Yes, this sentence has been revised: ‘With better understanding of management practices to optimize production, lucerne pasture may become more prevalent due to its drought resistance, tolerance to grazing and high quality [7]’ (lines 59-61).
Line 64: Remove "(Medicago sativa)".
- Removed (line 64).
Lines 138–140: Please add brief description of Bevenutti's and Ison's methods to understand how to determine pasture intake.
- The description of the method to determine pasture intake has been expanded in section 2.3: “Pasture consumed at each fixed assessment point was calculated from the measured grazing depth and the top-down vertical distribution of the DM from section 2.2. Pasture intake was calculated as the average pasture consumed from the assessment points excluding the trampled and contaminated points.” (Lines 164 to 167).
Line 144: What is "mincing energy"?
- ‘Mincing energy’ has been removed (line 140).
Line 191–193: I believe that the replicate should be included in the GLM as a random factor.
- We have now specified this in the text:
“For the response (Y) variables, the independent experimental units were the replicates (each with two cows), with the replicates being fitted as random effects in the analyses” (lines 187 to 189).
Lines 235–259: Please double-check the position of Tables in the text. For example, the relationship between pre-grazing height and pasture content was shown in Table 3, but you referred Table 4 in line 236.
- In-text references to the tables and figures have been checked and corrected throughout the manuscript (lines 191-257).
Although, the most of the relationship between pre-grazing height and nutrient content were statistically significant, the slopes were rather low. For example, if the pasture height increase 10 cm, the CP content in the pasture decreases only 0.88%. Is there substantial meaning in this relationship? Even if the P-values show significance, you should consider its biological meaning.
- This is now included and discussed in section 4:
“This explains the variability of pasture quality as PGPH height increased. Despite the relationship between PGPH and nutrient content being statistically significant, the slopes of the relationships were rather low and probably biologically insignificant. Therefore, the observed increase in energy and nutrient intake with increasing PGPH was mainly driven by increasing pasture intake rather than the changes in nutrient content of the consumed pasture.” (lines 364 to 368)
Table 3 and 4: "ME intake" should be included in the titles, since all other items such as CP and NDF mean "content" in lucerne and total diet.
- Table titles have ME intake included (line 258 & 263).
Reviewer 2 Report
In this study, the height of alfalfa pre-grazing pasture was combined with the food intake and production performance of dairy cows. The experimental design was reasonable, and the results were logical, which had guiding significance for production practice. The author has carefully revised and explained the opinions of the experts in the initial evaluation.
Author Response
We appreciate the input that the reviewer has made to ensure that our manuscript meets the standards necessary to merit publication.
This manuscript is a resubmission of an earlier submission. The following is a list of the peer review reports and author responses from that submission.
Round 1
Reviewer 1 Report
This study includes some interesting findings and will be helpful for grazing management in the sub-tropical region using lucern as a pasture forage.
My main concern is the length of study period. This study conducted on April to May (less than one month). Plant structure of lucerne is different with its maturity (or seasons), even if the same plant height. In addition, you did not detect the effect of PGPH on milk production due to the short period of this study. So, I recommend that the authors should conduct an additional study and then confirm their results. Otherwise, your conclusions should be limited within your research condition.
Another concern is the roll of PUP in this study. Probably, one of main concept of this series of study is using the PUP as the pasture management index. However, the relationship between PGPH and PUP is unclear in the current manuscript. How the PGPH affects PUP or how the PGPH works under the PUP grazing strategy. Please explain more details about this relation in the introduction and discussion section.
In addition, the current form of the Tables and Figures do not enough for publication. They lack many important information such as SE and CI, and some Tables are really confusing (for the details see below).
Details:
Title: “Lucern” or “lucern pasture” should be included in the title.
Introduction
In some region of the world, lucern may be used as a common pasture forage; however, in many regions, it not so common. It is helpful to provide an appropriate reason to use lucern as a pasture forage in this study.
The authors described three hypotheses in the objective, but it lacks mechanism or reasons that support these hypotheses. For example, why pasture intake would have quadratic relationship with PGPH and why PMR allocation would not affect total DMI? Add brief reasons for these hypotheses.
Lines 102–103: What is the statistical units? In this study, each treatment has two replications. If the statistical unit is the replicates, I believe that sample size is small.
Line 108: What is “feed pad”? It is not common term.
Line 133: Please define NDF and ADF more precisely. NDF and ADF exclude or include ash? Did you use a-amylase when you determine NDF?
Line 147: How many transects did you prepared in each replicate? In addition, add the length of the transect.
Lines 150–151: Explain how to define and measure the trampled, contaminated and un-grazable areas. These areas tightly relate to PUP.
Lines 158–159: Did you taken milk sample every day during the study period? You should clearly state this in this section.
Lines 167–168: You mentioned that you use step-down selection method to find the appropriate model. If so, provide the information of the full statistical model at the beginning. P-value is a criterion, but coefficient is also an important criterion.
Lines 218, 219, 226 and 230: I am confused. These items such as ME, NDF and ADF indicate contents or intake?
Line 251: I did not find figure 7 and 8.
Line 252: Figure 4 instead of figure 6?
Lines 315–324: Mezzalira et al (2014) used grassed (Cynodon sp. and Avena strigose) in their study. Plant structure and morphology are different between grass and legume species. This difference would lead to different result between the studies. Thus, you should include species difference in the discussion. In addition, linear and quadratic relationship between DMI and PGPH depends on the actual pasture height. Therefore, actual pasture height in the studies should be included in the discussion. Moreover, the current discussion lacks the reason why the result did not fit your hypothesis. This should be also included in the revised version.
The second hypothesis should be discussed. In the original manuscript, you just mentioned the results.
Line 332: ME content or intake?
Lines 346–347: This statement is quite ambiguous. You mean the result is due to the short duration of the study, as indicated in lines 360–361.
Tables:
1) All abbreviations in the tables should be explained in the footnotes.
2) “Pasture category”: you did not use this word in the main text, and should explain in the text if you use it.
3) The tables only indicated mean values of the items, but indicating SD, SE or CI are helpful to understand the results.
4) I do not understand the bottom half of the tables. Probably, those mean the effect of PMR on each item including pasture offered, grazing depth and so on. In my opinion, those variables should be summarized in another table and add other variables such as RMSE and significance of each coefficient.
5) Table 4 and 5: If these values mean nutrient “intake”, the values should be expressed as g DM/cow or kg DM/cows. If these values mean the nutritional contents of the pasture, you should not express these as “nutrient intake”.
Figures: Statistical results should be included in the figures.
Reviewer 2 Report
This paper reports the effect of pre-grazing pasture heights and supplementation of partially mixed rations on intake and performance of grazing dairy cows. The results are interesting and provide some new knowledge about grazing management of dairy cattle. However, the authors need to address the following concerns:
1. The authors designed a 3 (pasture heights) ´ 2 (ration supplementation) factorial experiment in dairy cows (Table 1). However, their statistical analyses did not reveal the main effect of the two factors and their interactions; rather, linear regressions were developed instead. Although the pasture height failed to reach the designated target, it still remains to be one of the two factors required to be statistically tested; they should not be treated as ‘explanatory variables’ (L180), however. Therefore, two-way ANOVA should be carried out for the observations so that the main effect of each factor as well as their interactions can be statistically clarified.
2. The authors conclude that ‘39 cm’ of pre-grazing pasture heights above ground is optimal for pasture intake by dairy cows (L34, L321, L323, and L371). However, dairy cows forage in 3 dimensions; therefore, pasture intake by the cows can never be associated solely with one dimension – the height of pasture herbages; the other two dimensions in term of ‘pasture production’ (or vegetation density) expressed in kg of DM per hectare, must accompany the pasture height at the same time. Thus, the optimal pasture height should always be coupled with pasture production; otherwise, the conclusion can never be conclusive.
3. For the experimental design, the authors allocated two replicates of 2 cows each into each treatment (Table 1). The necessity to adopt such a design is doubtful. Why not treat each animal individual as a replicate so that the sample size (n) will be 4 for each treatment? An animal trial looks awkward when its n = 2, although treating a group of animals as the replicate may reduce the intra-group variability.
4. Pasture intake is an essential parameter for a feeding trial with grazing animals. The authors measured the pasture DM intake by grazing cows. However, the results of two-way ANOVA for these observations are not presented in any Tables or Figures, thus hindering the reader to associate the performance of the cows with their intake, and to assess whether the substitution effect of the partially mixed ration supplementation have occurred. Furthermore, the reader cannot trace down the method adopted by the authors for measuring pasture DM intake by the cows because the entry of the reference (L410) listed at the end of the text is incomplete. However, one can deduce from relevant descriptions in the text that the authors estimated pasture intake of the grazing cows by an indirect method. The authors have to indicate the accuracy of such an indirect method in comparisons with in vivo pasture intake by the animals.
5. The values in Tables 3 to 5 are raw observations of replicates; the value of means for treatments, along with P values of main effects and their interactions for each measurement, is expected, instead. Moreover, nutrient intake should be expressed as absolute values rather than as ‘% DM’ as shown in Table captions.
6. Numerous stylistic errors are found in the text. For example, the full name of ‘PMR’ should be defined in title (L3); ‘… [2] .’ (L48) should be ‘… [2].’; ‘… [2,3]’ (L51) should be ‘… [2,3].’; ‘[4,5] however …’ should be ‘[4,5]; however …’; ‘Ison, et al. [7] …’ (L57) should be ‘Ison et al. [7] …’; ‘The diet quality (Metabolisable Energy, ME MJ/cow.day; Crude Protein, CP % DM; Neutral Detergent Fibre, NDF % DM; Acid Detergent Fibre, ADF % DM; Lignin % DM; Starch, % DM and Sugars, % DM)’ (L215-217) should read as ‘The diet quality [metabolisable energy (ME), MJ/cow.day; crude protein (CP), %DM; neutral detergent fibre (NDF), %DM; acid detergent fibre (ADF), %DM; lignin, %DM; starch, %DM; and sugars, %DM)’; certain words are missing in the statement ‘Furthermore, the area grazed increased as PGPH increased which further explains the higher pasture intakes achieved when PGPH The reason for the larger grazing area is unclear.’ (L304-306); many errors occur in entries of References section. These errors are unacceptable for a scientific paper, and jeopardize the academic value of a research study.